# CUT&Tag for efficient epigenomic profiling of small samples and single cells

Hatice S. Kaya-Okur[1,2], Steven J. Wu[1,3], Christine A. Codomo[1,2], Erica S. Pledger[1], Terri D. Bryson[1,2], Jorja G. Henikoff[1], Kami Ahmad [1] & Steven Henikoff[1,2]

Many chromatin features play critical roles in regulating gene expression. A complete understanding of gene regulation will require the mapping of specific chromatin features in small samples of cells at high resolution. Here we describe Cleavage Under Targets and Tagmentation (CUT&Tag), an enzyme-tethering strategy that provides efficient high-resolution sequencing libraries for profiling diverse chromatin components. In CUT&Tag, a chromatin protein is bound in situ by a specific antibody, which then tethers a protein A-Tn5 transposase fusion protein. Activation of the transposase efficiently generates fragment libraries with high resolution and exceptionally low background. All steps from live cells to sequencing-ready libraries can be performed in a single tube on the benchtop or a microwell in a high-throughput pipeline, and the entire procedure can be performed in one day. We demonstrate the utility of CUT&Tag by profiling histone modifications, RNA Polymerase II and transcription factors on low cell numbers and single cells.

[1] Basic Sciences Division, Fred Hutchinson Cancer Research Center, 1100N. FairviewAve, Seattle, WA 98109, USA. [2] Howard Hughes Medical Institute, Fred Hutchinson Cancer Research Center, Seattle, WA 98109, USA. [3] Molecular Engineering & Sciences Institute, University of Washington, Seattle, WA 98195, USA. Correspondence and requests for materials should be addressed to K.A. (email: kahmad@fredhutch.org) or to S.H. (email: steveh@fhcrc.org)

The advent of massively parallel sequencing and the dramatic reduction in cost per base has fueled a genomics revolution, however, the full promise of epigenomic profiling has lagged owing to limitations in methodologies used for mapping chromatin fragments to the genome[1]. Chromatin immunoprecipitation with sequencing (ChIP-seq) and its variations[2–5] suffer from low signals, high backgrounds and epitope masking due to cross-linking, and low yields require large numbers of cells[2,6]. Alternatives to ChIP include enzyme-tethering methods for unfixed cells, such as DamID[7], ChEC-seq[8], and CUT&RUN[9,10], where a specific protein of interest is targeted in situ and then profiled genome-wide. For example, CUT&RUN, which is based on Laemmli's Chromatin ImmunoCleavage (ChIC) strategy[11], maps a chromatin protein by successive binding of a specific antibody, and then tethering a Protein A/Micrococcal Nuclease (pA-MNase) fusion protein in permeabilized cells without cross-linking[9]. MNase is activated by addition of calcium, and fragments are released into the supernatant for extraction of DNA, library preparation and paired-end sequencing. CUT&RUN provides base-pair resolution of specific chromatin components with background levels that are much lower than with ChIP-seq, dramatically reducing the cost of genome-wide profiling. Although CUT&RUN can generate high-quality data from as few as 100–1000 cells, it must be followed by DNA end polishing and adapter ligation to prepare sequencing libraries, which increases the time, cost and effort of the overall procedure. Moreover, the release of MNase-cleaved fragments into the supernatant with CUT&RUN is not well-suited for application to single-cell platforms[12,13].

Here we overcome the limitations of ChIP-seq and CUT&RUN using a transposome that consists of a hyperactive Tn5 transposase[14,15]—Protein A (pA-Tn5) fusion protein loaded with sequencing adapters. Tethering in situ followed by activation of pA-Tn5 results in factor-targeted tagmentation, generating fragments ready for PCR enrichment and DNA sequencing. Beginning with live cells, Cleavage Under Targets and Tagmentation (CUT&Tag) provides amplified sequence-ready libraries in a day on the bench top or in a high-throughput format. We show that a variety of chromatin components can be profiled with exceptionally low backgrounds using low cell numbers and even single cells. This easy, low-cost method will empower epigenetic studies in diverse areas of biological research.

## Results

**Efficient profiling of nucleosomes and RNAPII with CUT&Tag.** To implement chromatin profiling by tagmentation (Fig. 1a), we incubated intact permeabilized human K562 cells with an antibody to lysine-27-trimethylation of the histone H3 tail (H3K27me3), an abundant histone modification that marks silenced chromatin regions. We incubated cells with a secondary anti-rabbit antibody to increase the local concentration of antibody bound on chromatin sites, then incubated cells with an excess of pA-Tn5 fusion protein pre-loaded with sequencing adapters to tether the enzyme at antibody-bound sites in the nucleus. The transposome has inherent affinity for exposed DNA[16,17], and so we washed cells under stringent conditions to remove un-tethered pA-Tn5. We then activated the transposome by addition of $Mg^{++}$, integrating adapters spanning sites of H3K27me3-containing nucleosomes. Finally, fragment libraries were enriched from purified DNA and pooled for multiplex paired-end sequencing on an Illumina HiSeq flow-cell. The entire protocol manipulates all steps in a single reaction tube (Fig. 1b), where permeabilized cells are first mixed with an antibody, and then immobilized on Concanavalin A-coated paramagnetic beads, allowing magnetic handling of the cells in all successive wash and

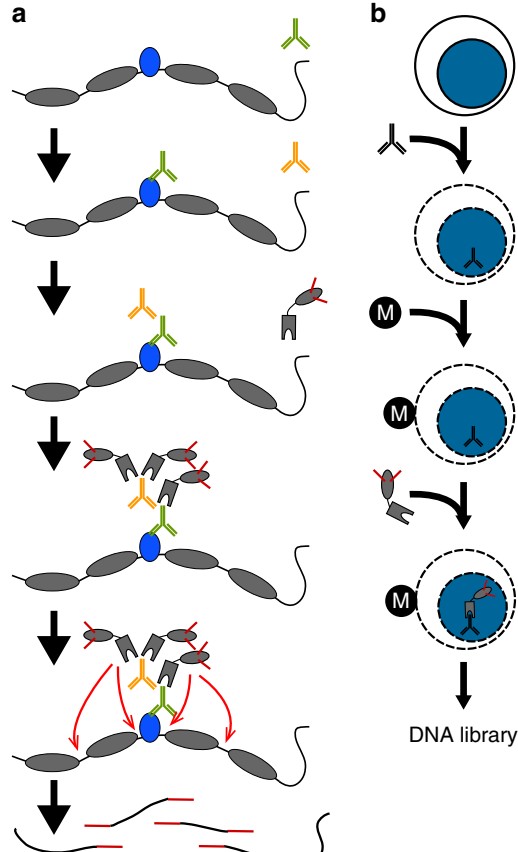

**Fig. 1** In situ tethering for CUT&Tag chromatin profiling. **a** The steps in CUT&Tag. Added antibody (green) binds to the target chromatin protein (blue) between nucleosomes (gray ovals) in the genome, and the excess is washed away. A second antibody (orange) is added and enhances tethering of pA-Tn5 transposome (gray boxes) at antibody-bound sites. After washing away excess transposome, addition of $Mg^{++}$ activates the transposome and integrates adapters (red) at chromatin protein binding sites. After DNA purification genomic fragments with adapters at both ends are enriched by PCR. **b** CUT&Tag is performed on a solid support. Unfixed cells or nuclei (blue) are permeabilized and mixed with antibody to a target chromatin protein. After addition and binding of cells to Concanavilin A-coated magnetic beads (M), all further steps are performed in the same reaction tube with magnetic capture between washes and incubations, including pA-Tn5 tethering, integration, and DNA purification

reagent incubation steps. For standardization between experiments, we used the small amount of tracer genomic DNA derived from the *E. coli* during transposase protein production to normalize sample read counts in lieu of the heterologous spike-in DNA that is recommended for CUT&RUN[9] (see Methods section and Supplementary Fig. 1a).

Display of ~8 million reads mapped to the human genome assembly shows a clear pattern of large chromatin domains marked by H3K27me3 (Fig. 2a). We also obtained profiles for H3K4me1 and H3K4me2 histone modifications, which mark active chromatin sites. In contrast, incubation of cells with a non-specific IgG antibody, which measures untethered integration of adapters, produced very sparse landscapes (Fig. 2a). To assess the signal-to-noise of CUT&Tag relative to other methods we compared it with profiling generated by CUT&RUN[18] and by ChIP-seq[19] for the same H3K27me3 rabbit monoclonal antibody in K562 cells. To directly compare the three techniques, we set the read depth of each dataset to 8 million reads each. Landscapes for each of the three methods are similar, but background noise

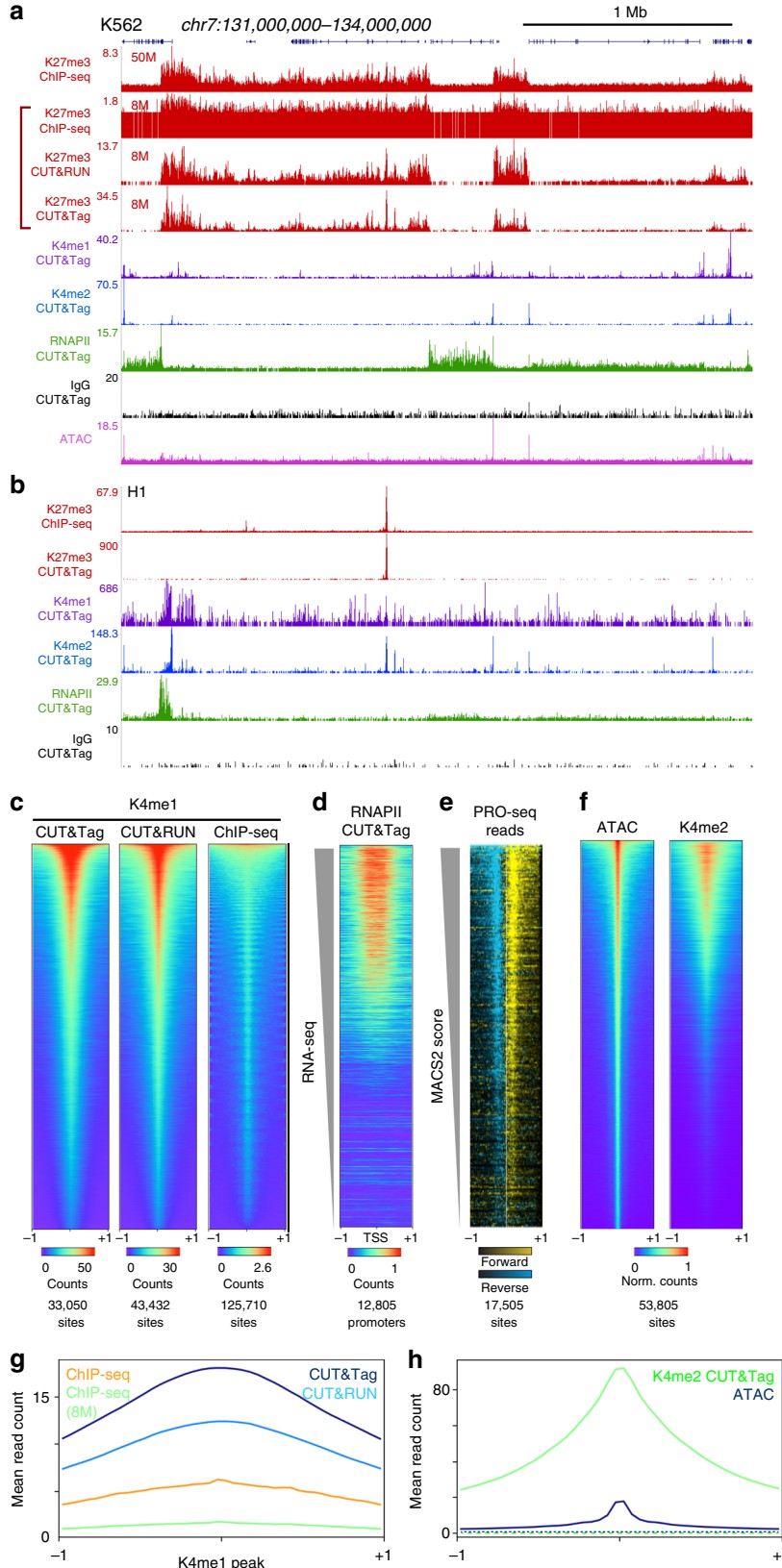

dominates in ChIP-seq datasets (Fig. 2a), and it is thus appears that ChIP-seq will require substantially greater read depth to distinguish chromatin features from background. In contrast, both CUT&RUN and CUT&Tag profiles have extremely low background noise levels. As expected, very different profiles were seen in the same region for a different human cell type, H1 embryonic stem (H1 ES) cells (Fig. 2b). To more quantitatively compare signal and noise levels in each method, we generated heatmaps around genomic sites called from H3K4me1 modification profiling for each method, where the same antibody had been

**Fig. 2** CUT&Tag for histone modification profiling and RNAPII. **a** Representative chromatin landscapes across a 3 Mb segment of the human genome generated by the indicated method. For H3K27me3, we downsampled ChIP-seq and CUT&RUN datasets to the same total mapped read counts as CUT&Tag for direct comparison. The high background in downsampled ChIP-seq is from singleton reads distributed across the genome. **b** Same as **a** except for H1 ES cells. **c** Comparison of profiling methods for the H3K4me1 histone modification in K562 cells. The same antibody was used in all experiments. Peaks were called and ordered for each dataset using MACS2. Each dataset was downsampled to the same read depth for comparison and plotted on their called peaks. Color intensities are scaled to the maximum read count at peaks in each dataset. **d** Detection of gene activity by RNAPII CUT&Tag. Gene promoters were ordered by associated RNA-seq counts (gray wedge) and read counts from RNAPII S2/5p CUT&Tag were plotted on these sites. **e** Active RNAPII is enriched at RNAPII CUT&Tag peaks. Peaks were called from RNAPII S2/5p CUT&Tag and ordered using MACS2 (gray wedge). PRO-seq reads were displayed onto these positions for (+) strand reads (yellow) and (−) strand reads (blue). **f** Comparison of ATAC-seq and H3K4me2 CUT&Tag profiling in K562 cells. Peaks were called on ATAC-seq data and heat maps were produced as in **c**. The top and bottom 2.5% of peaks were discarded to remove outliers. **g** Metaplot comparison of H3K4me1 histone modification signal in CUT&RUN, CUT&Tag, and ChIP-seq in K562 cells, averaged at the top 10,000 peaks detected by MACS2 in ChIP-seq data. Profiling with the same antibody was compared at the downsampled read depths of 8 million mapped reads for all three methods (blue, cyan, and green), and for 40 million mapped reads (orange) from ChIP-seq. **h** Metaplot comparison of ATAC-seq and H3K4me2 CUT&Tag profiling in K562 cells. 53,805 peaks were called on ATAC-seq data using MACS2, and read counts from each method were averaged across the intervals. The top and bottom 2.5% of peaks were discarded to remove outliers. Read counts at 17,000 randomly-chosen intervals for each dataset are displayed as dotted lines. Source data are available in the Source Data file

used. After sampling each dataset to 8 million reads for comparison, we found that CUT&Tag for this histone modification shows moderately higher signals compared to CUT&RUN throughout the list of sites (Fig. 2c). Both methods have low backgrounds around the sites. In contrast, ChIP-seq signal has a very narrow dynamic range that is ~1/20 of the CUT&Tag signal range, and much weaker signals across the majority of sites. To quantitatively compare methods, we displayed the average read counts for CUT&Tag, CUT&RUN and ChIP-seq datasets for the H3K4me1 histone mark around the top 10,000 peaks defined by MACS2 on an H3K4me1 ChIP-seq dataset (Fig. 2g). We found that CUT&Tag profiling gives substantially more signal accumulation at these sites, implying that CUT&Tag will be most effective at distinguishing chromatin features with fewest reads.

The transcriptional status of genes and regulatory elements can be inferred from histone modification patterns, but gene expression is directly read out by profiling chromatin-bound RNA polymerase II (RNAPII). We used an antibody to the S2/S5-phosphorylation (S2/5p) forms of RNAPII, which distinguish engaged polymerase[20]. Landscapes show enrichment of RNAPII CUT&Tag reads at many genes (Fig. 2a, Supplementary Fig. 2a), and a promoter heatmap reveals that this enrichment is predominantly at the 5′ ends of active genes[21] (Fig. 2d). These results were confirmed by the observation of similar CUT&Tag patterns using antibodies to S2p, S5p and S7p forms of RNAPII (Supplementary Fig. 2a and Supplementary Fig. 3a, b).

To validate RNAPII CUT&Tag without relying upon annotations, which are typically based on mapping of processed transcripts, we chose transcriptional run-on data obtained with the base-pair-resolution PRO-seq technique, which provides direct mapping of RNAPII using a method that is unrelated to chromatin profiling[22]. PRO-seq maps the position of the 5′ end of engaged RNAPII as it is activated in situ, and is used to identify paused RNAPII just downstream of the transcriptional start site. Peaks were called from RNAPII S2/5p CUT&Tag and ordered using MACS2, and processed datasets from PRO-seq run-on for human K562 cells (SRA GSM1480327) were aligned to the peak calls. When ordered by RNAPII CUT&Tag MACS2 score, a close correspondence between PRO-seq occupancy and RNAPII-Ser2/5p CUT&Tag occupancy is seen (Fig. 2e). Similar heat maps were obtained using antibodies to S2p, S5p, and S7p phosphorylation of the RNAPII C-terminal domain (Supplementary Fig. 3c).

**CUT&Tag sensitively maps active sites in chromatin**. Replicates for profiling of H3K4me1 modification by CUT&Tag are highly similar, demonstrating the reproducibility of the method (Fig. 3a). We obtained similar reproducibility when we compared

H3K27me3 CUT&Tag replicates (Supplementary Fig. 2c). In previous experiments with CUT&RUN profiling, we found that H3K4me2 histone modification landscapes, which are associated with active promoters and enhancers, resemble ATAC-seq profiles[18]. We therefore performed CUT&Tag using an antibody to H3K4me2. An example of H3K4me2 CUT&Tag profiling to published ATAC-seq in K562 cells[23] is shown (Fig. 2a). We found high occupancies for H3K4me2 at strong ATAC-seq peaks (Fig. 2f), with much higher read counts (Fig. 2h), implying that H3K4me2 profiling captures the most prominent accessible chromatin sites in the genome with greater sensitivity.

To quantify the sensitivity of H3K4me2 CUT&Tag relative to H3K4me2 CUT&RUN[18], H3K4me2 ChIP-seq[19], and ATAC-seq[23], we downsampled reads from each method, and used MACS2 with default parameters to call peaks on each dataset. We then estimated the fraction of reads falling within the called peaks. We found that both CUT&RUN and CUT&Tag populate peaks more deeply than ChIP-seq or ATAC-seq, demonstrating that they have exceptionally low signal-to-noise (Fig. 3b). In addition, CUT&Tag more rapidly populates peaks at low sequencing depths, where ~2 million reads are equivalent to 8 million for CUT&RUN (or 20 million for ChIP-seq), demonstrating the exceptionally high efficiency of CUT&Tag. Of all the methods, only CUT&Tag reaches a fraction of 0.6 within peaks. Thus, with two histone modifications (H3K4me2 and H3K27me3), we segment the chromatin landscape into both active and silenced regions, even with relatively low sequencing depths.

**CUT&Tag simultaneously maps factor binding and accessible DNA**. To determine if we could use CUT&Tag for mapping transcription factor binding, we tested if pA-Tn5 tethered at transcription factors can be distinguished from accessible DNA sites in the genome. We used an antibody to the NPAT nuclear factor, a transcriptional coactivator of the replication-dependent histone genes, in CUT&Tag reactions. NPAT binds only ~80 accessible sites in the histone clusters on chromosome 1 and chromosome 6[24], thus we can compare true binding sites with accessible sites. In NPAT CUT&Tag profiles, ~99% of read counts accumulate at the promoters of the histone genes (Fig. 4a). By scoring sites for correspondence to published ATAC-seq data[23], we found that a smaller number of counts are distributed across accessible sites in the K562 genome (Fig. 4b). This probably results from some un-tethered pA-Tn5 binding to exposed DNA in situ, but it is straightforward to distinguish antibody-tethered sites from accessible sites by the vast difference in read coverage (Fig. 4c). Indeed, calling peaks by standard algorithms on NPAT CUT&Tag data generates a list of

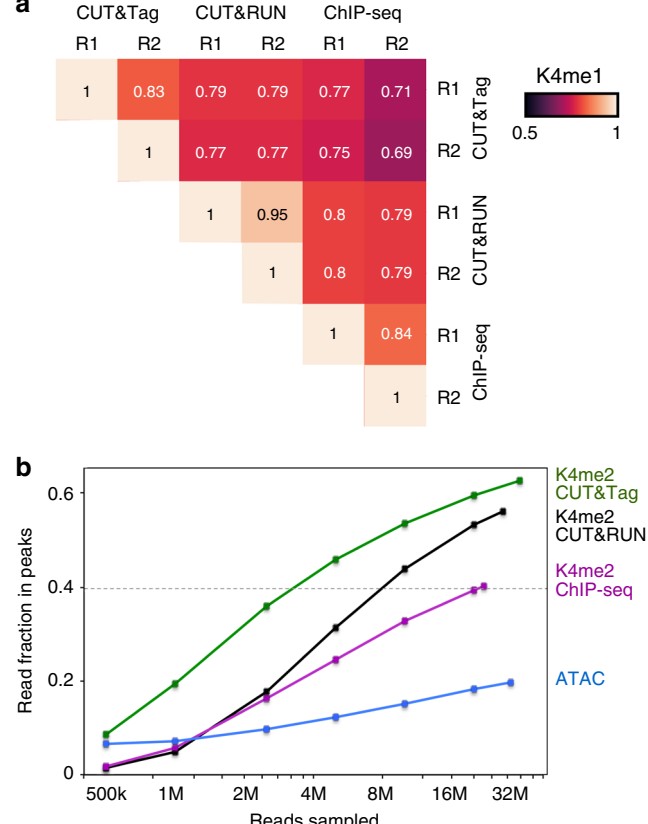

**Fig. 3** Reproducibility and efficiency of CUT&Tag. **a** Hierarchically clustered correlation matrix of CUT&Tag replicates (R1 and R2) and with CUT&RUN and ChIP-seq profiling for the H3K4me1 histone modification. The same antibody was used in all experiments. Pearson correlations were calculated using the $\log_2$-transformed values of read counts split into 500 bp bins across the genome. **b** Efficiency of peak-calling between methods. Mitochondrial reads were removed from datasets from each method profiling the H3K4me2 histone modification. The remaining read counts were downsampled to varying depths, and then used to call peaks using MACS2. The summed number of reads falling within called peaks in each dataset was plotted. Source data are available in the Source Data file

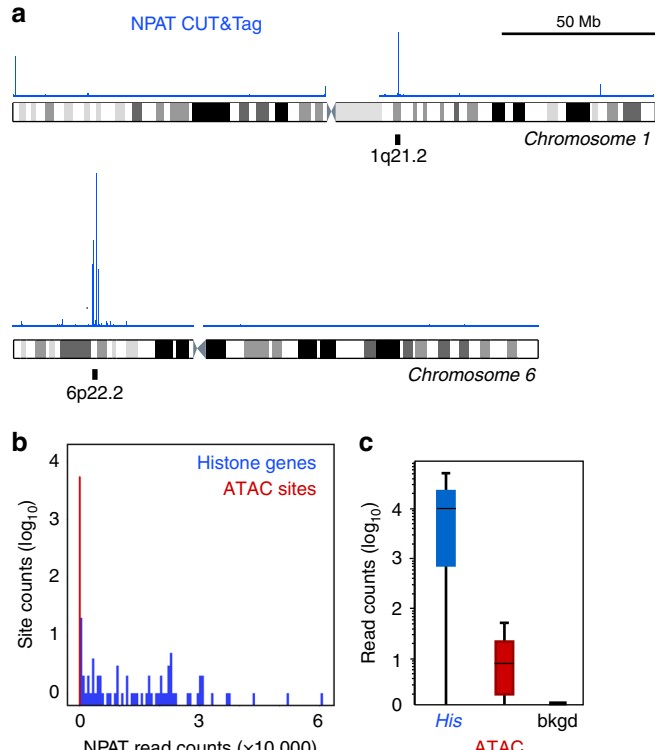

**Fig. 4** CUT&Tag profiling of the NPAT chromatin factor and chromatin accessibility. **a** Ideograms of chromosome 1 and chromosome 6 with the clusters of replication-dependent histone genes (6 genes at 1q21.2 and 55 genes at 6p22.2) indicated. An NPAT CUT&Tag profiling track is displayed over each chromosome. The major NPAT peaks fall at the histone genes. **b** Distribution of read counts in CUT&Tag profiling. Called accessible sites from ATAC-seq data were segregated into those at histone genes and other ATAC sites. Read counts from NPAT CUT&Tag were plotted for each category. **c** Boxplots of NPAT CUT&Tag signal at the promoters of replication-dependent histone genes (*His*), at other accessible sites called from ATAC-seq data, and at a random selection of genomic background sites (bkgd). Source data are available in the Source Data file

~9000 sites that includes histone gene promoters and 10% of ATAC-defined accessible sites (Supplementary Fig. 4). While this is only a fraction of the ~54,000 accessible sites defined in K562 cells, adjusting the threshold and stringency of NPAT peak calling may improve detection.

To test if CUT&Tag is tractable for profiling more abundant transcription factor binding sites, we profiled the CCCTC-binding factor (CTCF) DNA-binding protein. For these experiments, we varied the stringency of wash buffers to assess displacement of transcription factors from chromatin. Under low-salt and medium-salt concentration conditions we observed read counts at CTCF sites detected by CUT&RUN and by ChIP-seq (Fig. 5a), but with additional minor peaks (Supplementary Fig. 2a). These additional peaks suggest that un-tethered pA-Tn5 contributes to coverage in these experiments. To determine if true CTCF binding sites could be distinguished from accessible features by read depth, we compared the CUT&Tag read count at high-confidence CTCF sites (defined by peak-calling on CUT&RUN data[18]) to the CUT&Tag read count at accessible sites (defined by peak-calling on ATAC-seq data[23]). We found that these two distributions of read counts overlap, but that of accessible sites is lower than that of CTCF sites (Fig. 5b). Based

solely on read depth, we discriminate ~5600 CTCF bound sites with a 1% false discovery rate. Comparing motif enrichment in these two classes demonstrates that the high signals correspond to CTCF motifs ($E$-value $= 2.1 \times 10^{-69}$), and the low signals do not.

We assessed the resolution of the CUT&Tag procedure by plotting the ends of reads centered on CTCF binding sites. This shows that CUT&Tag protects a "footprint" spanning 80 bp directly over the CTCF motif (Fig. 5c). While the segment protected from Tn5 integration is larger than the ~45 bp protected from MNase in CUT&RUN[9], this indicates that the tethered transposase produces high resolution maps of factor binding sites. Similar footprints were obtained using different salt concentration washes, although 300–500 mM salt concentrations resulted in somewhat reduced signal-to-noise (Fig. 5c). The high resolution of CUT&Tag provides structural details of individual sites. For example, superposition of CTCF, H3K4me1, H3K4me2, H3K4me3, and ATAC mapping at a representative site reveals the relationship between accessible DNA, CTCF binding, and modified neighboring nucleosomes (Fig. 5d).

**CUT&Tag profiles low cell number samples and single cells.** ChIP requires substantial cellular material, limiting its application for experimental and clinical samples. However, we and others have previously demonstrated that tethered profiling strategies

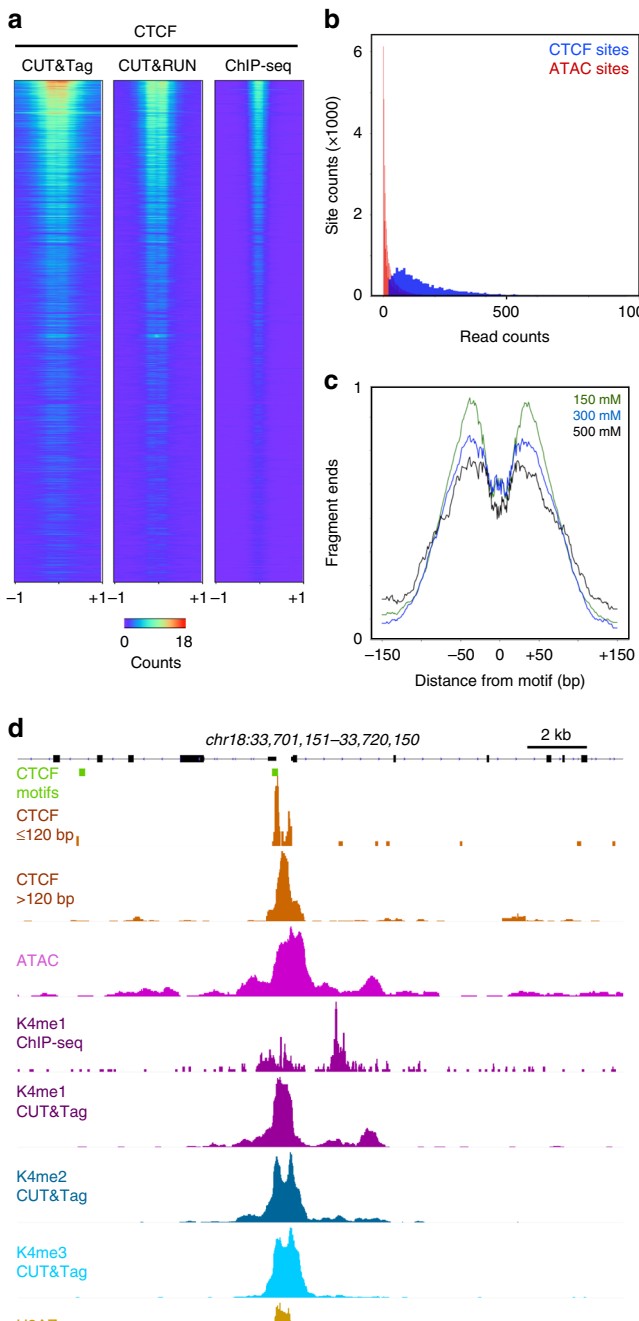

**Fig. 5** CUT&Tag profiling of the CTCF DNA-binding protein. **a** Comparison of methods for CTCF mapping. CTCF motifs in the genome were ranked by *e*-value, datasets from each method were downsampled to the same read depth, and then read counts were plotted on the fixed order of sites. **b** Distribution of read counts in CTCF CUT&Tag profiling. Sites were called from CUT&RUN profiling (blue) and at non-overlapping accessible sites (red, called from ATAC-seq). Read counts from CTCF CUT&Tag were plotted for each category. **c** Resolution of CUT&Tag. Mean plots of fragment end positions from CTCF CUT&Tag centered over CTCF motifs in called peaks. Three different NaCl concentrations were used in wash buffers during and after pA-Tn5 tethering. Data are represented as a fraction of the maximum signal within the interval. **d** Resolved structure of a CTCF binding site. The promoter of the *SLC39A6* gene on chromosome 18 shows the chromatin features around a CTCF-bound site. Source data are available in the Source Data file

like CUT&RUN have sufficient sensitivity that profiling small cell numbers routinely becomes feasible[9,25]. Signal improvements in CUT&Tag suggest that this method may work even more efficiently with limited samples. We first tested CUT&Tag for the H3K27me3 modification across a ~1500× range of material, from 100,000 down to 60 cells. We observed very similar high-quality chromatin profiles from all experiments (Supplementary Fig. 1b), demonstrating that high data quality is still maintained with low input material. Analyzing sample and tracer DNA in these CUT&Tag series revealed that sequencing yield is proportional to the number of cells (Supplementary Fig. 1a).

CUT&Tag has the advantage that the entire reaction from antibody binding to adapter integration occurs within intact cells. The transposase and chromatin fragments remain bound together[15,26], and thus fragmented DNA is retained within each nucleus. We developed a simple strategy to generate chromatin profiles of individual cells, which we term single-cell CUT&Tag (scCUT&Tag) (Fig. 6a). We performed scCUT&Tag to the H3K27me3 modification on a bulk population of K562 cells, but with gentle centrifugation between steps instead of Concanavalin A magnetic beads. After integration, we used a Takara ICELL8 nano-dispensing system to aliquot single cells into nanowells of a 5184 well chip, identifying the nanowells that contained one and only one cell by imaging the chip. We then performed PCR enrichment of libraries in each passing nanowell using two indexed primers, and finally pooled all enriched libraries from the chip for Illumina deep sequencing to high redundancy to assess the sampling and coverage in each cell (Supplementary Fig. 6a). Libraries from each well are distinguished by unique combinations of the two indices.

The aggregate of single cell chromatin profiling closely matched profiles generated in bulk samples (Fig. 6b), with high correlations (Pearson's $r = 0.89$, Supplementary Fig. 6b). Individual cells were ranked by the genome-wide number of reads, and the unique fragments are displayed in tracks for each cell. Strikingly, the majority of reads from individual cells fall within H3K27me3 blocks defined in bulk profiling, indicating high recovery in single cell chromatin profiling (Fig. 6b). A second replicate of H3K27me3 scCUT&Tag demonstrated the reproducibility of single cell profiling. Similarly, single cell profiling of the H3K4me2 modification recapitulates genomic landscapes of accessible and active chromatin (Fig. 6c). A significant fraction of reads in single cells fall within defined active and silenced chromatin features (Fig. 6d, e).

The breadth of chromatin features—from ~5 nucleosomes for H3K4me2 to hundreds in H3K27me3 domains—assists the detection of chromatin features even with sparse sampling from individual cells. To assess if chromatin features in individual cells could be used to distinguish cell types, we performed scCUT&Tag to the H3K27me3 modification in H1 cells. Again, we found that a high fraction of reads fell within domains defined by bulk profiling (Fig. 6e), with high correlations between bulk and aggregated single cell data (Pearson's $r = 0.85$, Supplementary Fig. 6b). Comparing a 2 Mb region encompassing the *HoxB* domain reveals clear histone methylation in single cell tracks specifically in H1 cells, while this region is depleted in K562 cells (Fig. 6f). These genome-wide patterns are sufficient to discriminate single H1 cells from K562 cells with high efficiency (Supplementary Fig. 6c, d). The small fraction of K562 cells that are mis-called have the sparsest read coverage. Thus, chromatin profiling provides a method to discriminate single cell types.

## Discussion
Chromatin profiling by CUT&Tag efficiently reveals regulatory information in genomes. In contrast to RNA-seq[27], which only

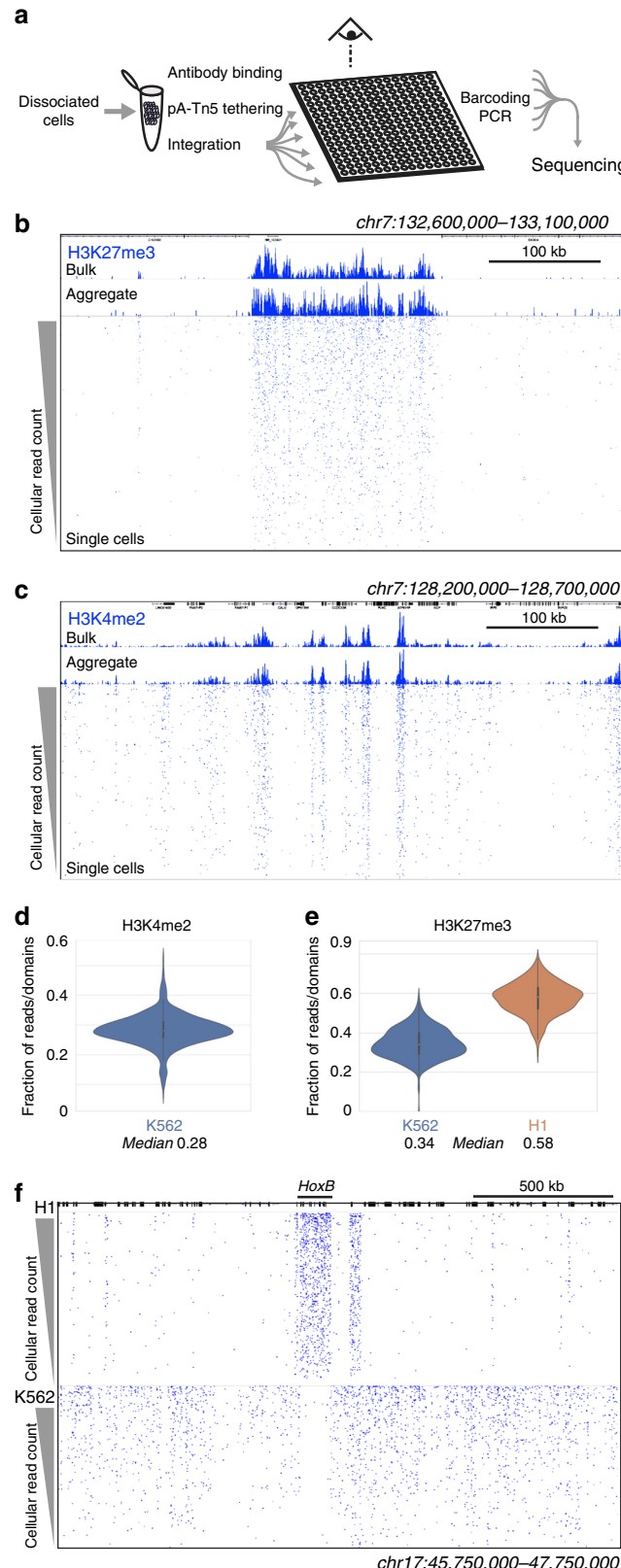

**Fig. 6** Chromatin profiling of individual cells. **a** Single cell CUT&Tag (scCUT&Tag) processing. All steps from antibody incubations through adapter tagmentation are done on a population of permeabilized unfixed cells. Individual cells are then dispensed into nanowells of a Takara ICELL8 chip. After verifying nanowells with single cells by microscopy, combinations of two indexed barcoded primers are added to each well and fragment libraries are enriched by PCR. Libraries from the chip are pooled for multiplex sequencing. **b** A chromatin landscape across a 500 kb segment of the human genome is shown for H3K27me3 CUT&Tag on K562 cells. Tracks from bulk CUT&Tag, aggregated scCUT&Tag, and for 956 single cells are shown. Single cells are ordered by total read counts in each cell. **c** A chromatin landscape across a 500 kb segment of the human genome for H3K4me2 CUT&Tag on K562 cells. Tracks from bulk CUT&Tag, aggregated scCUT&Tag, and for 808 single cells are shown. Single cells are ordered by total read counts in each cell. **d** Fraction of reads in single K562 cells falling within called active peaks for the H3K4me2 histone modification using stringent criteria. Narrow peaks were called using MACS2 on bulk profiling data, and reads from scCUT&Tag were assigned to those peaks. **e** Fraction of reads in single K562 and H1 cells falling within called silenced domains for the H3K27me3 histone modification. Domains were called using SEACR on bulk profiling data for each cell type, and reads from scCUT&Tag were assigned to those domains. **f** Comparison of chromatin landscapes in H1 and K562 single cells across a 2 Mb segment including the *HoxB* locus. Four hundred and seventy-nine single cells of each type were ordered by total read counts. Source data are available in the Source Data file

have been extensively used in model cell line systems, the vagaries of crosslinking and fragmenting chromatin have limited chromatin profiling by ChIP-seq to an artisan technique where each experiment requires optimization. Likewise, a recently described alternative cross-linked chromatin profiling method, ChIL-seq[28], requires many more steps than CUT&Tag and requires 3–4 days to perform all of the steps. In contrast, the CUT&Tag procedure, like CUT&RUN, is an unfixed in situ method, and is easily implemented in a standardized approach. This, combined with the cost-effectiveness of CUT&Tag, makes it appropriate for a high-throughput pipeline that can be implemented in a core facility[18]. It is conceivable that diverse users may provide their mixture of cells and antibody and receive processed deep sequencing files in just days. Since the first step in high-throughput CUT&Tag is antibody incubation at 4 °C, samples can be accumulated overnight in a facility and then loaded together onto a 96-well plate for robotic handling, as we previously demonstrated for AutoCUT&RUN[18]. With efficient use of reagents and better signal-to-noise, CUT&Tag requires even fewer reads per sample than AutoCUT&RUN, which is already much cheaper than commercial exome sequencing. While the ease and low cost of this pipeline is appealing, the primary virtue of automated chromatin profiling is the minimization of batch and handling effects, and thus maximum reproducibility. Such aspects are critical for clinical assays and testing for chromatin-targeting drugs.

We have shown that CUT&Tag provides high-quality single-cell profiles using the ICELL8 nano-dispensation system[12], which allows for imaging prior to reagent addition and PCR. Likewise, CUT&Tag should be suitable for the 10× Genomics encapsulation system[13] by adaptation of their recently announced ATAC-seq single-cell protocol[29]. Adaptability to high-throughput single-cell platforms is possible for CUT&Tag because adapters are added in bulk, whereas previous single-cell adaptations of antibody-based profiling methods, including ChIP-seq[30], ChIL-seq[28], and CUT&RUN[25] require reactions to be performed after cells are

measures expressed genes, chromatin profiling has the unique advantage of identifying silenced regions, which is a key aspect of establishing cell fates in development. Although methods like ATAC-seq map accessible and factor-bound sites[17], the specific chromatin proteins bound at these sites must be inferred from motif or chromatin profiling data. While ChIP-based methods

separated. The distinct distributions of low-level untargeted accessible DNA sites and high-level CTCF-bound sites in CUT&Tag datasets suggests that by modeling the two expected underlying distributions, true binding sites can be distinguished from accessible DNA sites without using other data. An advantage of this strategy is that the statistical distinction between true binding sites and accessible features allows characterization of two chromatin features in the same experiment, where accessible DNA sites can be annotated as well as binding sites for the targeted factor. This parsing out of the low-level ATAC-seq background from the strong targeted CUT&Tag signal makes possible de novo "multi-OMIC" CUT&Tag[31]. In the future, we expect that barcoding of adapters[26] will allow for multiple epitopes to be simultaneously profiled in single cells in large numbers, maximizing the utility of single-cell epigenomic profiling for studies of development and disease.

## Methods

**Biological materials**. Human K562 cells were purchased from ATCC (Manassas, VA, Catalog #CCL-243) and cultured following the supplier's protocol. H1 ES cells were obtained from WiCell (Cat#WA01-lot#WB35186). We used the following antibodies: Guinea Pig anti-Rabbit IgG (Heavy & Light Chain) antibody (Antibodies-Online ABIN101961). H3K27me3 (Cell Signaling Technology, 9733, Lot 14), H3K27ac (Millipore, MABE647), H3K4me1 (Abcam, ab8895), H3K4me2 (Upstate 07–030, Lot 26335), H3K4me3 (Active Motif, 39159), PolSer2P, PolSer5P, PolSer2+5P, PolSer7P (Cell Signaling Technology, Rpb1 CTD Antibody Sampler Kit, 54020), CTCF (Millipore 07–729), NPAT (Thermo Fisher Scientific, PA5–66839 ALX-215-065-1), and Sox2 (Abcam, ab92494).

**Transposome preparation**. Using the pTXB1-Tn5[15] expression vector, sequences downstream of lac operator were replaced with an efficient ribosome binding site, three tandem FLAG epitope tags and two IgG binding domains of staphylococcal protein A, which were PCR amplified from the pK19pA-MN vector[11]. The C-terminus of Protein A was separated from the transposase by a 26 residue flexible linker peptide composed of DDDKEF(GGGGS)$_4$. The pTXB1-Tn5 plasmid was a gift from Rickard Sandberg (Addgene plasmid # 60240) and the pK19pA-MN plasmid was a gift from Ulrich Laemmli (available through Addgene, plasmid # 86973). The 3XFlag-pA-Tn5-Fl plasmid (Addgene plasmid # 124601) was transformed into C3013 cells (NEB) following the manufacturer's protocol. Each colony tested was inoculated into 3 mL LB medium and growth was continued at 37 °C for 4 h. That culture was used to start a 400 mL culture in 100 µg/mL carbenicillin containing LB medium (as it is more stable than ampicillin) and incubated on a shaker until it reached O.D. ~0.6, whereupon it was chilled on ice for 30 min. Fresh IPTG was added to 0.25 mM to induce expression, and the culture was incubated at 18 °C on a shaker overnight. The culture was collected by centrifugation at 10,000 rpm, 4 °C for 30 min. The bacterial pellet was frozen in a dry ice-ethanol bath and stored at −70 °C. Protein purification was performed as previously described[15] with minor modifications. Briefly, a frozen pellet was resuspended in 40 mL chilled HEGX Buffer (20 mM HEPES-KOH at pH 7.2, 0.8 M NaCl, 1 mM EDTA, 10% glycerol, 0.2% Triton X-100) including 1× Roche Complete EDTA-free protease inhibitor tablets. The lysate was sonicated 10 times for 45 s at a 50% duty cycle with output level 7 while keeping the sample chilled and holding on ice between cycles. The sonicated lysate was centrifuged at 10,000 rpm in a Fiberlite rotor at 4 °C for 30 min. A 2.5 mL aliquot of chitin slurry resin (NEB, S6651S) was packed into each of two disposable columns (Bio-rad 7321010). Columns were washed with 20 mL of HEGX Buffer. The soluble fraction was added to the chitin resin slowly, then incubated on a rotator at 4 °C overnight. The unbound soluble fraction was drained and the columns were rinsed with 20 mL HEGX and washed thoroughly with 20 mL HEGX containing Roche Complete EDTA-free protease inhibitor tablets. The chitin slurry was transferred to a 15 mL conical tube and resuspended in elution buffer (10 mL HEGX with 100 mM DTT). The tube was placed on rotator at 4 °C for ~48 h. The eluate was collected and dialyzed twice in 800 mL 2X Tn5 Dialysis Buffer (100 mM HEPES-KOH pH 7.2, 0.2 M NaCl, 0.2 mM EDTA, 2 mM DTT, 0.2% Triton X-100, 20% Glycerol). The dialyzed protein solution was concentrated using an Amicon Ultra-4 Centrifugal Filter Units 30 K (Millipore UFC803024), and sterile glycerol was added to make a final 50% glycerol stock of the purified protein.

To generate the pA-Tn5 adapter transposome, 16 µL of a 100 µM equimolar mixture of preannealed Tn5MEDS-A and Tn5MEDS-B oligonucleotides[15] were mixed with 100 µL of 5.5 µM pA-Tn5 fusion protein. The mixture was incubated on a rotating platform for 1 h at room temperature and then stored at −20 °C. The complex is stable at room temperature, with no detectable loss of potency after 10 days on the benchtop (Supplementary Fig. 5A), and without noticeable loss of data quality (Supplementary Fig. 5B). Unexpectedly, this extended room temperature incubation resulted in a 1–2 order-of-magnitude increase in the number of tagmented *E. coli* fragments (Supplementary Fig. 5C), which can be used as a calibration standard within a series using a constant amount of pA-Tn5. This

observation suggests that the *E. coli* DNA that co-purifies with pA-Tn5 is subject to tagmentation both during room temperature incubation and during tagmentation in situ, where the dramatic increase seen with pre-incubation results from subsequent trapping of tagmented pA-Tn5-bound DNA within the cell. In support of this interpretation, we note that *E. coli* carry-over DNA suitable for calibration is also released by pA-MNase in CUT&RUN reactions during digestion[32]. A likely explanation for the trapping of these different fusion protein-bound DNAs within cells is that the protein-protein interaction domains of Protein A that are specific for IgG bind non-specifically to cellular proteins, whereupon addition of divalent cation results in MNase digestion and release (pA-MNase) or tagmentation (pA-Tn5).

**CUT&Tag for bench-top application**. Cells were harvested, counted and centrifuged for 3 min at 600×*g* at room temperature. Aliquots of cells (60–500,000 cells), were washed twice in 1.5 mL Wash Buffer (20 mM HEPES pH 7.5; 150 mM NaCl; 0.5 mM Spermidine; 1× Protease inhibitor cocktail) by gentle pipetting. Concanavalin A coated magnetic beads (Bangs Laboratories) were prepared as described[9] and 10 µL of activated beads were added per sample and incubated at RT for 15 min. We observed that binding cells to beads at this step increases binding efficiency. The unbound supernatant was removed and bead-bound cells were resuspended in 50–100 µL Dig-wash Buffer (20 mM HEPES pH 7.5; 150 mM NaCl; 0.5 mM Spermidine; 1× Protease inhibitor cocktail; 0.05% Digitonin) containing 2 mM EDTA and a 1:50 dilution of the appropriate primary antibody. Primary antibody incubation was performed on a rotating platform for 2 h at room temperature (RT) or overnight at 4 °C. The primary antibody was removed by placing the tube on the magnet stand to clear and pulling off all of the liquid. To increase the number of Protein A binding sites for each bound antibody, an appropriate secondary antibody (such as Guinea Pig anti-Rabbit IgG antibody for a rabbit primary antibody) was diluted 1:50 in 50–100 µL of Dig-Wash buffer and cells were incubated at RT for 30 min. Cells were washed using the magnet stand 2–3× for 5 min in 0.2–1 mL Dig-Wash buffer to remove unbound antibodies. A 1:200 dilution of pA-Tn5 adapter complex (~0.04 µM) was prepared in Dig-med Buffer (0.05% Digitonin, 20 mM HEPES, pH 7.5, 300 mM NaCl, 0.5 mM Spermidine, 1× Protease inhibitor cocktail). After removing the liquid on the magnet stand, 50–100 µL was added to the cells with gentle vortexing, which was incubated with pA-Tn5 at RT for 1 h. Cells were washed 2–3× for 5 min in 0.2–1 mL Dig-med Buffer to remove unbound pA-Tn5 protein. Next, cells were resuspended in 50–100 µL Tagmentation buffer (10 mM MgCl$_2$ in Dig-med Buffer) and incubated at 37 °C for 1 h. To stop tagmentation, 2.25 µL of 0.5 M EDTA, 2.75 µL of 10% SDS and 0.5 µL of 20 mg/mL Proteinase K was added to 50 µL of sample, which was incubated at 55 °C for 30 min or overnight at 37 °C, and then at 70 °C for 20 min to inactivate Proteinase K. To extract the DNA, 122 µL Ampure XP beads were added to each tube with vortexing, quickly spun and held 5 min. Tubes were placed on a magnet stand to clear, then the liquid was carefully withdrawn. Without disturbing the beads, beads were washed twice in 1 mL 80% ethanol. After allowing to dry ~5 min, 30–40 µL of 10 mM Tris pH 8 was added, the tubes were vortexed, quickly spun and allowed to sit for 5 min. Tubes were placed on a magnet stand and the liquid was withdrawn to a fresh tube.

To amplify libraries, 21 µL DNA was mixed with 2 µL of a universal i5 and a uniquely barcoded i7 primer[33], using a different barcode for each sample. A volume of 25 µL NEBNext HiFi 2× PCR Master mix was added and mixed. The sample was placed in a Thermocycler with a heated lid using the following cycling conditions: 72 °C for 5 min (gap filling); 98 °C for 30 s; 14 cycles of 98 °C for 10 s and 63 °C for 30 s; final extension at 72 °C for 1 min and hold at 8 °C. Post-PCR clean-up was performed by adding 1.1× volume of Ampure XP beads (Beckman Counter), and libraries were incubated with beads for 15 min at RT, washed twice gently in 80% ethanol, and eluted in 30 µL 10 mM Tris pH 8.0.

A detailed, step-by-step protocol can be found at
https://www.protocols.io/view/bench-top-cut-amp-tag-wnufdew/abstract

**High-throughput CUT&Tag**. For high-throughput 96-well microplate application, cells were first permeabilized and incubated with the primary antibodies before binding to beads. Two biological replicates of human K562 and H1 ES cells were washed twice with Wash Buffer, resuspended in Dig-wash buffer with 2 mM EDTA and arrayed in a 96-well plate. Permeabilization before antibody incubation varied from 1 to 5 h. Then, dilutions of appropriate antibodies were added (making final antibody concentrations 1:50) as duplicates. Cells were incubated with primary antibodies overnight. The next day, 10 µL of activated Concanavalin A coated magnetic beads were added to each sample, mixed gently and incubated at room temperature for 10 min. The plate was placed on a magnetic plate holder and supernatants were discarded. Appropriate secondary antibodies were prepared as 1:50 dilutions in Dig-wash and added to each well. Cells were washed three times with Dig-wash and then incubated with 1:200 dilution of pA-Tn5 adapter complex in Dig-med buffer at RT for 1 h. Cells were washed 3× for 5 min in Dig-med Buffer and resuspended in 50 µL Tagmentation buffer and incubated at 37 °C for 1 h. To stop tagmentation, 2.25 µL of 0.5 M EDTA, 2.75 µL of 10% SDS and 0.5 µL of 20 mg/mL Proteinase K was added to the sample, which was incubated at 55 °C for 30 min and then at 70 °C for 20 min to inactivate Proteinase K. Samples were held at 4 °C overnight until ready to continue. A 1.1× volume of AMPure XP beads was added to each well, vortexed and incubated at room temperature for

10–15 min. The plate was placed on a magnet and unbound liquid was removed. Beads were gently rinsed twice with 80% ethanol, and DNA was eluted with 35 µL of 10 mM Tris-HCl pH 8. 30 µL of eluted DNA was amplified by PCR as described above.

**DNA sequencing and data processing**. The size distribution of libraries was determined by Agilent 4200 TapeStation analysis, and libraries were mixed to achieve equal representation as desired aiming for a final concentration as recommended by the manufacturer. Paired-end Illumina sequencing was performed on the barcoded libraries following the manufacturer's instructions. Paired-end reads were aligned using Bowtie2 version 2.2.5 with options: –local–very-sensitive-local–no-unal–no-mixed–no-discordant–phred33 -I 10 -X 700. Because of the very low background with CUT&Tag, typically 3 million paired-end reads suffice for nucleosome modifications, even for the human genome. For maximum economy, up to 96 barcoded samples per 2-lane flow cell can be pooled for 25 × 25 bp sequencing. For peak calling, parameters used were macs2 callpeak—t input_file –p 1e-5 –f BEDPE/BED(Paired End vs. Single End sequencing data) –keep-dup all –n out_name.

**Single-cell CUT&Tag**. Approximately 50,000 exponentially growing K562 cells were processed by centrifugation between buffer and reagent exchanges in low-retention tubes throughout. Centrifugations were performed at 600×g for 3 min in a swinging bucket rotor for the initial wash and incubation steps, and then at 300×g for 3 min after pA-Tn5 binding. Cells were collected and washed twice with 1 mL Wash Buffer (20 mM HEPES, pH 7.5; 150 mM NaCl; 0.5 mM Spermidine, 1× Protease inhibitor cocktail) at room temperature. Nuclei were isolated by permeabilizing cells in NP40-Digitonin Wash Buffer (0.01% NP40, 0.01% Digitonin in wash buffer) and resuspended in 1 mL of NP40-Digitonin Wash buffer with 2 mM EDTA. Antibody was added at a 1:50 dilution and incubated on a rotator at 4 °C overnight. Permeabilized cells were then rinsed once with NP40-Digitonin Wash buffer and incubated with anti-Rabbit IgG antibody (1:50 dilution) in 1 mL of NP40-Digitonin Wash buffer on a rotator at room temperature for 30 min. Nuclei were then washed 3× for 5 min in 1 mL NP40-Digitonin Wash buffer to remove unbound antibodies. For pA-Tn5 binding, a 1:100 dilution of pA-Tn5 adapter complex was prepared in 1 mL NP40-Dig-med-buffer (0.01% NP40, 0.01% Digitonin, 20 mM HEPES, pH 7.5, 300 mM NaCl, 0.5 mM Spermidine, 1× Protease inhibitor cocktail), and permeabilized cells were incubated with the pA-Tn5 adapter complex on a rotator at RT for 1 h. Cells were washed 3× for 5 min in 1 mL NP40-Dig-med-buffer to remove excess pA-Tn5 protein. Cells were resuspended in 150 µL Tagmentation buffer (10 mM MgCl2 in NP40-Dig-med-buffer) and incubated at 37 °C for 1 h. Tagmentation was stopped by adding 50 µL of 4× Stop Buffer (40.4 mM EDTA and 2 mg/mL DAPI) and the sample was held on ice for 30 min.

The SMARTer ICELL8 single-cell system (Takara Bio USA, Cat. #640000) was used to array single cells as described for scATAC-seq[12]. DAPI-stained nuclei were visualized under the microscope and if there were clumps, they were strained through 10 micron cell strainers. Cells were counted using a hematocytometer and diluted at 28 cells/µL in 0.5× PBS and 1× Second Diluent (Takara Bio USA, Cat. # 640196). Cells were loaded to a source loading plate. Control wells containing 0.5× PBS (25 µL) and fiducial mix (25 µL) (Takara Bio USA, Cat. #640196) were also included in the source loading plate. Using the ICELL8 MultiSample NanoDispenser (MSND) FLA program, cells were dispensed into a SMARTer ICELL8 350 v chip (Takara Bio USA, Cat. # 640019) at 35 nanoliter per well. After cell dispense was complete, chips were sealed with the imaging film (Takara Bio USA, Cat. #640109) and centrifuged at 400×g for 5 min at room temperature and imaged using the ICELL8 imaging station (Takara Bio USA). Images were analyzed using automated microscopy image analysis software (CellSelect, Takara Bio USA). Since cells were stained only with DAPI, they were propidium iodide negative, so that permeabilized cells would not be excluded by default software settings. Additional single cells were manually selected for dispensing using a manual triaging procedure. Immediately following imaging, the filter file, which notes single-cell containing wells and control wells, was generated. We typically obtained ~1000 single cells per chip. All of the following reagents were added to the selected set of wells which contained single cells. To index the whole chip, 72 i5 and 72 i7 unique indices[33] were dispensed at 7.32 µM using ICELL8 MSND FLA program using the index 1 and index 2 filtered dispense tool respectively at 35 nanoliter per well. NEBNext High-Fidelity 2X PCR Master Mix (NEB, M0541L) was dispensed twice using the ICELL8 MSND Single Cell/TCR program for the filtered dispense tool at 50 nL per well. The chip was sealed and centrifuged at 2250×g at 4 °C for at least 5 min after each dispense. The chip was sealed with a TE Sealing film (Takara Bio USA, Cat. #640109) and on-chip PCR was performed using a SMARTer ICELL8 Thermal Cycler (Takara Bio USA) as follows: 5 min at 72 °C and 2 min at 98 °C followed by 15 cycles of 10 s at 98 °C, 30 s at 60 °C, and 5 s at 72 °C, with a final extension at 72 °C for 1 min. PCR products were collected by centrifugation at ~2250×g for 20 min using the supplied SMARTer ICELL8 Collection Kit (Takara Bio USA, Cat.#640048).

Pooled libraries were purified using Ampure XP beads (Beckman Counter) in a 1:1.1 ratio. Briefly, libraries were incubated with beads for 15 min at RT, washed twice in 80% ethanol, and eluted in 10 mM Tris pH 8.0. Paired-end 25 × 8 × 8 × 25 bp Illumina sequencing was performed on the pooled barcoded libraries following the manufacturer's instructions. Paired-end reads were aligned using Bowtie2 version 2.2.5 with options:–local–very-sensitive-local–no-unal–no-mixed–no-discordant–phred33 -I 10 -X 700.

**Reporting summary**. Further information on research design is available in the Nature Research Reporting Summary linked to this article.

### Data availability
Publically available datasets analyzed in this work are available in Supplementary Note 1. All sequencing data generated in this study have been deposited in GEO under accession GSE124557. The 3XFlag-pA-Tn5-Fl plasmid has been deposited with Addgene (#124601). Source data for the figures can be found in the Source Data file. All other data are available from the authors upon reasonable request.

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

## Acknowledgements
We thank Tayler Hentges and Aaron Hernandez for technical support, and Matt Fitzgibbon and Michael Meers for helpful discussions on data analysis. We also thank all members of the Henikoff lab for valuable discussions. This work was supported by the Howard Hughes Medical Institute, grants from the National Institutes of Health (R01 GM 108699 and 4DN TCPA A093) and the Fred Hutch Summer Undergraduate Research Program.

## Author contributions
H.S.K. and S.H. designed and performed all experiments. C.A.C and E.S.P. and T.D.B. assisted with the experiments. H.S.K., S.J.W., J.G.H., K.A., and S.H. developed algorithms and analyzed the data. H.S.K, K.A., and S.H. wrote the manuscript.

## Additional information

**Competing interests:** The authors declare no competing interests.

