## [Peer Review File · Nature Communications]

Reviewers' comments:

Reviewer #1 (Remarks to the Author):

In their manuscript 'CUT&Tag for efficient epigenomic profiling of small samples and single cells', the authors describe a novel technique to obtain highly specific profiles of chromatin modifications and transcription factors. CUT&Tag is based on a proteinA-Tn5 transposase fusion protein that's recruited to genomic loci using specific antibodies. This is an exciting method which improves upon their previous method ('Cut&Run'). It appears to improve signal and noise compared to ChIP-seq and allows for high-throughput and single cell applications.

The method seems to constitute a valuable addition both for basic research and clinical applications. However, the manuscript requires revisions to make this point clearer and to make it possible to assess the value of this method independently.

Assessment of CUT&Tag and comparison to other methods is almost exclusively qualitative and performance of the single cell protocol is not assessed at all. The authors need to describe some experimental and most analyses in more detail.

Related to Figure 1

The figure legend mentions the orange antibody that's used to increase tethering, presumably a secondary Ab targeting the constant region of the primary Ab. While the material and method section does mention secondary antibodies, this should also be mentioned in the text.

Related to Figure 2

The figure would be easier to interpret if genes would be shown in 'squish' mode as this makes clearer where genes start and end. All tracks should be shown with the same scale or axes should be labelled.

"..and ChIP-seq thus requires significantly more read depth to distinguish chromatin features from this background". This assessment would be better based on proportion of reads in a set of defined positive regions and background regions.

Figure 2c: this is not a quantitative comparison. Instead, the authors could compare the difference in the mean for the distributions of read counts at the center of the regions. This comparison also suffers from the use of different sets of sites in each plot. It would be more appropriate to use a single set of regions on which to compare the signal of all 3 methods.

Figure 2 f: “It is apparent that H3K4me2 profiling captures accessible chromatin sites in the genome, with greater sensitivity at lower read depths (Fig. 2f).” It is not clear what this sentence means in the context of figure 2 f. Relative to what is H3K4me2 capturing open chromatin with higher sensitivity. What was the expectation for this analysis?

“Thus CUT&Tag is most effective at distinguishing chromatin features with fewest reads.”

I would like to see a more quantitative assessment before making this statement. This is not to say that the presented data suggesting otherwise –the analyses presented are just not quantitative enough to make that case.

Related to Figure 3a

The comparison in this figure would benefit from the exclusion of the IgG controls. It seems that the main message is the correlation between replicates of a given technique and those of another technique. Compared to IgG all of them are very similar and the more subtle differences cannot be appreciated.

Related to Figure 3b

The figure legend reads very different than the description in the text. It is not clear what the authors have done. The text suggest that peaks were called and proportion of subsampled reads within peaks was measured. I’m puzzled that this fraction would change with read sampling. However, the legend and excel file suggest that reads were down-sampled and for each sampling the peaks were called and proportion of reads in peaks were assessed.

Legend and text need to be adjusted and method section needs to contain a detailed description of the analysis procedure.

“Thus, with two histone modifications (H3K4me2 and H3K27me3) and relatively low sequencing depths, we capture most of the regulatory information in chromatin landscapes for both active and silenced chromatin.”

It is not clear how the authors arrive at this conclusion: What is the definition of ‘most of the regulatory information in chromatin landscapes for both active and silenced chromatin’?

The section header: “CUT&Tag simultaneously maps transcription factor binding sites and accessible DNA” and the following sentence in the discussion “This parsing out the low-level ATAC-seq background from the strong targeted CUT&Tag signal, makes possible de novo “multi-OMIC” CUT&Tag30.” are confusing. It is not clear how the authors would manage this: I agree with their assertion that sites associated with motifs for the targeted factor (e.g. CTCF) are enriched to a degree that makes it easy to distinguish them from background (e.g. sites that are accessible but without motif in the same cell line). However, the ability to distinguish TF sites from unbound, accessible background sites says nothing about the ability in turn to effectively identify those accessible sites and distinguish them from non-accessible sites using the same data.

Related to Figure 5d:

This figure demonstrates that CTCF binding detected by CUT&Tag coincides with accessibility and chromatin modifications associated with active regulatory regions. It’s not obvious that this is a particularly high-resolution view of the data or that the assertion ‘Resolved structure of a CTCF binding sites’ is warranted. Is it possible to see the motif + footprint at CTCF sites when zooming in? Tracks should be shown with the same scale or axes should be labelled.

Related to figure 6:

Single cell CUT&Tag is an exciting application and the combination with the iCell8 system allows for high throughput automated library preparation. However, the authors do not provide enough data to assess the quality of their single cell data.

QC methods for single cell methods should be performed and presented. Such as:

How many reads fall on each cell barcode (mean and standard deviation)?

What’s the read duplication rate per cell?

What’s the correlation between aggregated profiles and bulk data for the same mark?

Using peaks/regions called in bulk data what’s the proportion of reads in single cell falling into peak regions?

Do aggregate profiles at different regions (e.g. TSS) show the expected distribution in single cells?

How similar are the profiles in single cells (e.g. Jaccard index based on coverage at bulk peak regions)

Supplemental Fig 4 – labelling of x axis unclear (presumable ‘rank’ of the cell based on number of unique reads?)

Reviewer #2 (Remarks to the Author):

Kaya-Okur et al. “CUT&Tag for efficient epigenomic profiling of small samples and single cells”

The authors have developed a novel method, CUT&Tag, to map chromatin modifications and RNApolIII. This method is an improvement on their previously published CUT&RUN due to its applicability for single cell analysis, as the authors indicate. The key improvement relies on the Tn5 transposase’s ability to circumvent the inefficient adaptor ligation step (which is needed in CUT&RUN and other methods), instead directly load the barcoded adaptor on the genomic DNA fragment. This is a very interesting and highly promising method for the wider research community, and since the data is convincing, I recommend publication of this work provided the authors address my concerns. Some key questions that need addressing are regarding specificity, in particular because of the claims around its usability in single cell.

1. The main concern of this reviewer is the false positive rate of signal detection in single cell experiments. If heterogeneity of a cell population is being analysed, it is quite important to know what confidence to give to the pattern in each single cell, and to be more specific, to each datapoint. Figure 6 show only a convenient window of the genome but more statistical analysis would be needed to establish what constitutes a “peak” in a single cell. How many neighbouring datapoints need to be analysed or pooled to raise above noise and consider it a “true” data? It would be actually quite helpful to, for example, sequence single cells of another cell type for H3K27me3 and perform statistical analyses comparing window sizes and confidence associated with it. Also, say take the top 500 cells from each cell type and mix a range of proportions in silico: what is the smallest % of cell subsets that can be “detected” in a heterogeneous population, in a statistically meaningful way? Is it 0.1%, is it 1% or is it 10%? This will help the research community evaluate the power and the limitations of this method.

2. Figure 2 data needs more quantitative analysis rather than showing only an example of a chromosome region (which is a good visualisation). Specifically, the same peak calling should be used on the 8M read data in ChIP-seq, CUT&RUN and CUT&Tag and statistically analyse the overlap between peaks in relation to ATAC-seq peaks. The expectation is that there will be more overlap with ATAC-seq of CUT&Tag, than ChIP-seq or CUT&RUN.
3. Protein A (pA-Tn5) fusion protein – will this (or the construct) commercially available?
4. “Concanavalin A 73 coated paramagnetic beads” – if this is essential in the protocol, does it mean that the binding efficiency will depend on the cell type? Are there some cells for which using this step will be a disadvantage?
5. The authors need to explain in more detail how the E. coli spike in experiment was performed in the manuscript. Can they provide an exact amount or proportion of E. coli DNA added? Was this added to the 100k cell and the mixture was serially diluted? Was a single sample serially diluted and separately the E. coli? Was this done in a single replicate? The Suppl Fig 1 legend or methods. “For standardization between experiments, we used the small amount 75 of tracer genomic DNA derived from the E. coli during transposase protein production to 76 normalize sample read counts in lieu of the heterologous spike-in DNA that is recommended for 77 CUT&RUN9 (Supplementary Fig. 1a).”
6. Annotate what the grey wedge is in Fig 2e
7. Were these two antibodies used separately? Which antibody was used in Fig2a? “We used an antibody to the S2/S5-phosphorylation (S2/5p) forms of 102 RNAPII, which distinguish engaged polymerase20”
8. Fig 3a, can numbers be displayed? Are the primary antibodies the same in Chip-seq and CUT&Tag? Can the authors show here the H3K27me3 correlations as well?
9. Fig 4. The authors need to show chr1 as well (in addition to Chr 6) and overlap with ATAC data.

10. Fig 4b is not clear, how was this analysis done, what was the width of the windows, was there zero overlap between histone genes and ATAC sites? What do they mean by “smaller number” in this sentence? How much percent exactly? “By scoring sites for correspondence to published ATAC-seq data²² 148 , we found that a smaller number of counts are distributed across accessible sites in the K562 genome” This is unclear, how many ATAC sites are expected in the NPAT regions? Can one calculate the rate of false positives from this data since NPAT sites are restricted? This would be very useful for the single cell analysis to estimate FDR (see point 1). The figure legend is not helping either: “b, Distribution of read counts in CUT&Tag profiling. Called accessible sites from ATAC-seq data 577 were segregated into those at histone genes and other ATAC sites. Read counts from NPAT 578 CUT&Tag were plotted for each category. ”

11. The authors need to be quantitative, which “minor peaks”? How many of these were detected? “Under low salt concentration conditions we observed read counts at 157 CTCF sites detected by CUT&RUN and by ChIP-seq (Fig. 5a), but with additional minor peaks 158 (Supplementary Fig. 2a). ”

12. Which cell line was used for the single cell sequencing? The methods says K562 but should be mentioned in the text and figure. To estimate the power of the method it would be important to test one of the antibodies (K27me3 or K4me3) in both cell lines presented in this paper. More statistical analysis would be very important to show to assess the heterogeneity in these populations.

13. Explain what “Carb360 containing LB medium” is.

14. It is not clear what “Paired-end 25x8x8x25 bp” means. Is the length of the sequencing read 25bp? Can the authors explain why they chose this read length? Is this only to maximise output and reduce cost or is there another reason?

Reviewer #1 (Remarks to the Author):

In their manuscript 'CUT&Tag for efficient epigenomic profiling of small samples and single cells', the authors describe a novel technique to obtain highly specific profiles of chromatin modifications and transcription factors. CUT&Tag is based on a proteinA-Tn5 transposase fusion protein that's recruited to genomic loci using specific antibodies. This is an exciting method which improves upon their previous method ('Cut&Run'). It appears to improve signal and noise compared to ChIP-seq and allows for high-throughput and single cell applications. The method seems to constitute a valuable addition both for basic research and clinical applications. However, the manuscript requires revisions to make this point clearer and to make it possible to assess the value of this method independently.

We thank Reviewer 1 for the positive overall evaluation of our manuscript

Assessment of CUT&Tag and comparison to other methods is almost exclusively qualitative and performance of the single cell protocol is not assessed at all. The authors need to describe some experimental and most analyses in more detail.

We have addressed each point with new analyses and/or revisions as requested.

Related to Figure 1

The figure legend mentions the orange antibody that's used to increase tethering, presumably a secondary Ab targeting the constant region of the primary Ab. While the material and method section does mention secondary antibodies, this should also be mentioned in the text.

We have added a sentence on p. 3 that describes adding secondary antibody.

Related to Figure 2

The figure would be easier to interpret if genes would be shown in 'squish' mode as this makes clearer where genes start and end. All tracks should be shown with the same scale or axes should be labelled.

We have now added labels for all browser track axes and improved the gene track for legibility.

"..and ChIP-seq thus requires significantly more read depth to distinguish chromatin features from this background". This assessment would be better based on proportion of reads in a set of defined positive regions and background regions.

We have adjusted to the text (p.3) to indicate that this appears to be the case from visual inspection of tracks. We now include the requested analysis in Figure 2g, where we selected the top 10,000 peaks defined by MACS2 of H3K4me1 ChIP-seq data, and displayed the average read counts for the three methods around these sites. A sentence is added on p.3. Note that it is not at all straightforward to define "true" positive and negative regions for a histone modification as there is no gold standard. However, the fact that CUT&Tag performs better than ChIP-seq even when we use high-scoring ChIP-seq reads as the standard, confirms our assertion.

Figure 2c: this is not a quantitative comparison. Instead, the authors could compare the difference in the mean for the distributions of read counts at the center of the regions. This comparison also suffers from the use of different sets of sites in each plot. It would be more appropriate to use a single set of regions on which to compare the signal of all 3 methods. **The requested analysis is now added as Figure 2g. We have retained Figure 2c because, in our opinion, this is a more "real world" analysis that displays how many peaks and what signal intensities are recovered in each method. A disadvantage of selecting a single set of regions is that data from one of the methods must be assumed to be true.**

However, as pointed out in response to the previous comment, CUT&Tag performs better than ChIP-seq even when using high-scoring ChIP-seq sites as the standard.

Figure 2 f: “It is apparent that H3K4me2 profiling captures accessible chromatin sites in the genome, with greater sensitivity at lower read depths (Fig. 2f).” It is not clear what this sentence means in the context of figure 2 f. Relative to what is H3K4me2 capturing open chromatin with higher sensitivity. What was the expectation for this analysis?

“Thus CUT&Tag is most effective at distinguishing chromatin features with fewest reads.” I would like to see a more quantitative assessment before making this statement. This is not to say that the presented data suggesting otherwise –the analyses presented are just not quantitative enough to make that case.

To address these two concerns, we have added Figure 2h, which displays the signal intensities around ATAC sites for both ATAC-seq and H3K4me2 CUT&Tag. ATAC and CUT&Tag datasets used here contain similar numbers of reads (~40 million mapped reads), and the higher signal observed at ATAC sites in H3K4me2 profiling is the basis of our conclusion that CUT&Tag profiling is more effective than ATAC-seq at detecting accessible sites. We have added text describing this analysis on p.4.

Related to Figure 3a

The comparison in this figure would benefit from the exclusion of the IgG controls. It seems that the main message is the correlation between replicates of a given technique and those of another technique. Compared to IgG all of them are very similar and the more subtle differences cannot be appreciated.

We have reformatted the figure as the reviewer suggested.

Related to Figure 3b

The figure legend reads very different than the description in the text. It is not clear what the authors have done. The text suggest that peaks were called and proportion of subsampled reads within peaks was measured. I’m puzzled that this fraction would change with read sampling. However, the legend and excel file suggest that reads were down-sampled and for each sampling the peaks were called and proportion of reads in peaks were assessed. Legend and text need to be adjusted and method section needs to contain a detailed description of the analysis procedure.

We have clarified the text on p.5 and the Figure legend. We have added a section in the Supplemental Information that contains detailed descriptions of the analysis procedures for each figure.

“Thus, with two histone modifications (H3K4me2 and H3K27me3) and relatively low sequencing depths, we capture most of the regulatory information in chromatin landscapes for both active and silenced chromatin.”

It is not clear how the authors arrive at this conclusion: What is the definition of ‘most of the regulatory information in chromatin landscapes for both active and silenced chromatin’?

We have changed the sentence to read: “Thus, with two histone modifications (H3K4me2 and H3K27me3), we segment the chromatin landscape into both active and silenced regions with relatively low sequencing depths.”

The section header: “CUT&Tag simultaneously maps transcription factor binding sites and accessible DNA” and the following sentence in the discussion “This parsing out the low-level ATAC-seq background from the strong targeted CUT&Tag signal, makes possible de novo

“multi-OMIC” CUT&Tag30.” are confusing. It is not clear how the authors would manage this: I agree with their assertion that sites associated with motifs for the targeted factor (e.g. CTCF) are enriched to a degree that makes it easy to distinguish them from background (e.g. sites that are accessible but without motif in the same cell line). However, the ability to distinguish TF sites from unbound, accessible background sites says nothing about the ability in turn to effectively identify those accessible sites and distinguish them from non-accessible sites using the same data.

We have added an additional analysis distinguishing factor-bound sites from accessible sites in a new Supplementary Figure 4 with accompanying text (p.5). We used MACS2 with a stringent threshold to detect peaks, and signal intensity at these peaks to distinguish NPAT-bound sites from accessible sites. Of 8689 peaks called on NPAT data that fall outside of the histone gene clusters, 5056 are ATAC sites that are strong enough to exceed the threshold, or ~10% of the ~54,000 previously called ATAC sites.

Related to Figure 5d:

This figure demonstrates that CTCF binding detected by CUT&Tag coincides with accessibility and chromatin modifications associated with active regulatory regions. It’s not obvious that this is a particularly high-resolution view of the data or that the assertion ‘Resolved structure of a CTCF binding sites’ is warranted. Is it possible to see the motif + footprint at CTCF sites when zooming in? Tracks should be shown with the same scale or axes should be labelled.

We have added the location of CTCF motifs and size-classed fragment data for CTCF CUT&Tag to this figure to make the point clear.

Related to figure 6:

Single cell CUT&Tag is an exciting application and the combination with the iCell8 system allows for high throughput automated library preparation. However, the authors do not provide enough data to assess the quality of their single cell data.

QC methods for single cell methods should be performed and presented. Such as:

How many reads fall on each cell barcode (mean and standard deviation)?

Supplementary Figure 4 gives the total number of unique fragments per cell, where the x-axis is the rank of the cell based on number of unique reads (new Supplementary figure 6a).

What’s the read duplication rate per cell?

We sequenced to near saturation and the read duplication rate/cell was ~98%. We calculated the number of unique reads using the Mark duplicates option of Picard tools, and report this estimated library size in Supplementary figure 6a.

What’s the correlation between aggregated profiles and bulk data for the same mark?

For H3K27me3 CUT&Tag Pearson’s $r = 0.89$ for K562 cells and $r = 0.85$ for H1 ES cells (Supplementary Figure 6b).

Using peaks/regions called in bulk data what’s the proportion of reads in single cell falling into peak regions?

We used SEACR (Meers *et al.* A streamlined protocol and analysis pipeline for CUT&RUN chromatin profiling, *bioRxiv* <https://www.biorxiv.org/content/10.1101/569129v1>) to call H3K27me3 broad domains and counted the number of reads falling into each domain, and we report the distribution in Figure 6e. The median fraction of reads in peaks for K562 cells is 0.34 and for H1 ES cells is 0.58.

Do aggregate profiles at different regions (e.g. TSS) show the expected distribution in single cells?

Yes, and we show representative examples in Figure 6c for TSSs and 6b for silencing domains.

How similar are the profiles in single cells (e.g. Jaccard index based on coverage at bulk peak regions)

We addressed this question by clustering (k=2) single-cell H3K27me3 datasets from a mixture of 479 H1 ES and 479 K562 cells for H3K27me3 (Supplementary Figure 6c). We find that all 479 H1 and 473 of 479 K562 cells were clustered correctly.

Supplemental Fig 4 – labelling of x axis unclear (presumable ‘rank’ of the cell based on number of unique reads?)

We agree and have changed the axis label accordingly (new Supplementary Figure 6a).

Reviewer #2 (Remarks to the Author):

Kaya-Okur et al. “CUT&Tag for efficient epigenomic profiling of small samples and single cells ”

The authors have developed a novel method, CUT&Tag, to map chromatin modifications and RNAPolIII. This method is an improvement on their previously published CUT&RUN due to its applicability for single cell analysis, as the authors indicate. The key improvement relies on the Tn5 transposase’s ability to circumvent the inefficient adaptor ligation step (which is needed in CUT&RUN and other methods), instead directly load the barcoded adaptor on the genomic DNA fragment. This is a very interesting and highly promising method for the wider research community, and since the data is convincing, I recommend publication of this work provided the authors address my concerns. Some key questions that need addressing are regarding specificity, in particular because of the claims around its usability in single cell.

We thank Reviewer 2 for the positive overall evaluation of our manuscript.

1. The main concern of this reviewer is the false positive rate of signal detection in single cell experiments. If heterogeneity of a cell population is being analysed, it is quite important to know what confidence to give to the pattern in each single cell, and to be more specific, to each datapoint. Figure 6 show only a convenient window of the genome but more statistical analysis would be needed to establish what constitutes a “peak” in a single cell. How many neighbouring datapoints need to be analysed or pooled to raise above noise and consider it a “true” data? It would be actually quite helpful to, for example, sequence single cells of another cell type for H3K27me3 and perform statistical analyses comparing window sizes and confidence associated with it. Also, say take the top 500 cells from each cell type and mix a range of proportions in silico: what is the smallest % of cell subsets that can be “detected” in a heterogeneous population, in a statistically meaningful way? Is it 0.1%, is it 1% or is it 10%? This will help the research community evaluate the power and the limitations of this method.

We thank Reviewer 2 for suggesting this experiment. We have performed scCUT&Tag on ~500 K562 and ~500 H1 cells on the same ICELL8 chip. When we stack the single-cell tracks centered over the HoxB locus, which is silenced in ES cells, we see that the HoxB cluster is strongly H3K27me3 trimethylated in H1 ESCs, essentially a reverse image of what is seen in K562 cells (below and new Figure 6d).

To evaluate this striking cell-type difference statistically, we performed k-means clustering on an equal mixture of H1 and K562 cells and obtained excellent separation as detailed in response to a similar request from Reviewer 1 (Supplementary Figure 6c).

2. Figure 2 data needs more quantitative analysis rather than showing only an example of a chromosome region (which is a good visualisation). Specifically, the same peak calling should be used on the 8M read data in ChIP-seq, CUT&RUN and CUT&Tag and statistically analyse the overlap between peaks in relation to ATAC-seq peaks. The expectation is that there will be more overlap with ATAC-seq of CUT&Tag, than ChIP-seq or CUT&RUN.

Reviewer 1 brought up similar issues with Figure 2c and Figure 2f. We have added quantitative comparisons as Figure 2g and Figure 2h.

3. Protein A (pA-Tn5) fusion protein – will this (or the construct) commercially available?

We appreciate that Reviewer 2 anticipates that our method will be of broad interest to the community, and this was our perception when we talked about this work in seminars and at meetings. As was the case for CUT&RUN, where there has been no commercial product available, we have assumed the task of distributing free samples of pA-Tn5 so that people can try CUT&Tag and provide feedback on our Protocols.io site. Accordingly, on March 6 2019, we released our submitted manuscript to the public on *bioRxiv* and the bench-top protocol on Protocols.io, and at the same time we sent a message to our >900 users on our CUT&RUN mailing list offering to send a free sample upon return of a FedEx label by e-mail, one per account. As shown in Supplementary Figure 5, the pA-Tn5 enzyme complex loses no significant activity at room temperature for at least 10 days, and this particular batch is sufficient for ~1000 aliquots, good for 50-200 CUT&Tag reactions for up to 100,000 cells.

We have also deposited the plasmid in Addgene, where it will be publically available (Addgene #124601). We expect that CUT&Tag commercial products based on our manuscript will appear to meet the demand.

4. “Concanavalin A 73 coated paramagnetic beads” – if this is essential in the protocol, does it mean that the binding efficiency will depend on the cell type? Are there some cells for which using this step will be a disadvantage?

We have used Concanavalin A beads extensively in our CUT&RUN protocols with many cell types and nuclei from mammalian, Drosophila, and budding yeast. We are aware in the literature of groups using these beads for CUT&RUN with plant cell nuclei (e.g., PMID: 30719569), thus they seem generally applicable. Note that magnetic handling is only convenient for bench-top and high-throughput CUT&Tag protocols, and in fact we do not use beads for the ICELL8 platform, which would preclude dispensing single cells for barcoding.

5. The authors need to explain in more detail how the *E. coli* spike in experiment was performed in the manuscript. Can they provide an exact amount or proportion of *E. coli* DNA added? Was this added to the 100k cell and the mixture was serially diluted? Was a single sample serially diluted and separately the *E. coli*? Was this done in a single replicate? The Suppl Fig 1 legend or methods. “For standardization between experiments, we used the small amount 75 of tracer genomic DNA derived from the *E. coli* during transposase protein production to 76 normalize sample read counts in lieu of the heterologous spike-in DNA that is recommended for 77 CUT&RUN9 (Supplementary Fig. 1a).”

Reviewer 2 seems to be under the impression that we spiked in the *E. coli* tracer DNA. We did not, and now we clarify the calibration procedure in the text and in the Methods. We have also added new Supplementary Figure 5, which more thoroughly describes the *E. coli* tracer DNA and the stability of the enzyme. *E. coli* DNA is a contaminant in each batch of pA-Tn5 we have produced from bacterial culture, and tagmentation occurs during the Mg⁺⁺ incubation step. Interestingly, by incubating pA-Tn5 complex for an extended period of time at room temperature, the amount of tagmented *E. coli* DNA that is recovered increases dramatically (Supplementary Figure 5c). We have reported a similar phenomenon with pA/MNase, where addition of calcium releases *E. coli* DNA fragments into the supernatant for extraction (Meers *et al.* A streamlined protocol and analysis pipeline for CUT&RUN chromatin profiling, *bioRxiv* <https://www.biorxiv.org/content/10.1101/569129v1>). A likely explanation for the trapping of these different fusion protein-bound DNAs within cells is that the protein-protein interaction domains of Protein A that are specific for IgG also bind non-specifically to cellular proteins with sufficient avidity to survive washing steps, whereupon addition of divalent cation results in MNase digestion and release (pA-MNase) or tagmentation (pA-Tn5). We now discuss this in the Methods section.

6. Annotate what the grey wedge is in Fig 2e

It is MACS2 counts and we have annotated this in the figure.

7. Were these two antibodies used separately? Which antibody was used in Fig2a? “We used an antibody to the S2/S5-phosphorylation (S2/5p) forms of 102 RNAPII, which distinguish engaged polymerase20”

This is a single antibody (CST13546S) that recognizes S2 and S5 phosphorylation of the RNAPII CTD.

8. Fig 3a, can numbers be displayed? Are the primary antibodies the same in Chip-seq and CUT&Tag? Can the authors show here the H3K27me3 correlations as well?

We have included the correlation numbers in the figure. We have added the statement on p.3 that the same antibody was used. We now provide a correlation matrix with numbers

for H3K27me3 where the same antibody was used for all three methods in Supplementary Figure 2c.

9. Fig 4. The authors need to show chr1 as well (in addition to Chr 6) and overlap with ATAC data.

We have added chromosome 1 to Figure 4a and have expanded analysis of overlap with ATAC data in Supplementary Figure 4.

10. Fig 4b is not clear, how was this analysis done, what was the width of the windows, was there zero overlap between histone genes and ATAC sites? What do they mean by “smaller number” in this sentence? How much percent exactly? “By scoring sites for correspondence to published ATAC-seq data²² 148 , we found that a smaller number of counts are distributed across accessible sites in the K562 genome” This is unclear, how many ATAC sites are expected in the NPAT regions?

We have included a thorough description of the analysis for each figure in a new Supplemental Information paragraph. We binned read counts within a +/- 100 bp region of each histone gene promoter in a NPAT CUT&Tag experiment. To determine read counts at hypersensitive sites, we repeated binning and counting for all ATAC-seq peaks that do not overlap with a histone gene promoter. To determine background, we selected 10,000 positions in the genome at random and counted reads at those sites. These read counts are plotted in the histogram.

Can one calculate the rate of false positives from this data since NPAT sites are restricted? This would be very useful for the single cell analysis to estimate FDR (see point 1). The figure legend is not helping either: “b, Distribution of read counts in CUT&Tag profiling. Called accessible sites from ATAC-seq data 577 were segregated into those at histone genes and other ATAC sites. Read counts from NPAT 578 CUT&Tag were plotted for each category. ”

The separation between NPAT true positives based on annotation and accessible sites is complicated by moderately strong NPAT binding sites that are not in a histone gene cluster. To circumvent this issue, we restricted the true positive NPAT set to peaks within the two histone gene clusters (on Chr1 and Chr6), and conservatively assumed that all other sites are accessible regions. Nevertheless, all of the highest read-count sites are in the histone gene clusters (Supplementary Figure 4).

11. The authors need to be quantitative, which “minor peaks”? How many of these were detected? “Under low salt concentration conditions we observed read counts at 157 CTCF sites detected by CUT&RUN and by ChIP-seq (Fig. 5a), but with additional minor peaks 158 (Supplementary Fig. 2a). ”

We have added an example of minor peaks in Supplementary Figure 2a, and indicate these with arrowheads. We have not counted how many minor peaks there were, because what to call a peak is subjective and depends on peak-calling parameters.

12. Which cell line was used for the single cell sequencing? The methods says K562 but should be mentioned in the text and figure. To estimate the power of the method it would be important to test one of the antibodies (K27me3 or K4me3) in both cell lines presented in this paper. More statistical analysis would be very important to show to assess the heterogeneity in these populations.

We have specified on p.7 and in the Figure 6 legend that K562 cells were used to develop scCUT&Tag. We have also added a major set of experiments profiling H3K27me3 in single H1 ES cells on the same chip as K562 cells as described above and in the new Supplementary Figure 6c. We scored single cells for reads falling in domains defined

from bulk profiling, and show that this efficiently discriminates the two cell types (Supplementary Figure 6c, d). We assess the performance of scCUT&Tag across varying read coverage/cell, and show that binning chromatin features is an effective way to handle even sparse chromatin profiling data.

13. Explain what “Carb360 containing LB medium” is.

We have corrected this to “carbenicillin”. Carbenicillin was used instead of ampicillin for overnight cultures because it is more stable.

14. It is not clear what “Paired-end 25x8x8x25 bp” means. Is the length of the sequencing read 25bp? Can the authors explain why they chose this read length? Is this only to maximise output and reduce cost or is there another reason?

Illumina 25x25 bp paired-end sequencing (8x8 refers to the indices) provides more than enough information from each fragment for unique mapping to the human genome. Longer read sequencing is required for identifying polymorphisms and mutations, but the task here is less demanding, which is to accurately map the ends of fragments.

Reviewer #1 (Remarks to the Author):

The authors addressed most of my concerns, however there are some points left that should be clarified before acceptance.

1. The analysis shown in figures 2f and 2h is still not clear to me. ATAC-seq identifies open chromatin regions and the authors use ATAC-seq peak calls to compare K4me2 Cut&Tag signal. They state that H3K4me2 detects open chromatin more 'effective' (rebuttal letter) or with 'greater sensitivity' (revised manuscript) than ATAC-seq. This does not seem to be supported by the data presented: The site-by-site comparisons of the heatmaps for ATAC-seq and K4me2 (Fig. 2f) Cut&Tag shows an ATAC-seq signal throughout. This is of course by design, since ATAC-seq peaks were used to select these regions. The K4me2 signal over the same regions on the right side shows that K4me2 shows higher signal at or around ~ 50% of these sites. Figure 2h shows aggregate plots of the data all regions used in 2f and demonstrates that the signal is much higher for K4me2.

Neither of these plots would lead to the conclusion that K4me2 is more effective than ATAC-seq at detecting open chromatin.

1. K4me2 does not seem to detect all open chromatin regions detected by ATAC-seq (2f).
2. The aggregate plot doesn't capture the distribution of the data across all regions. The signal is much higher than for ATAC-seq, but as apparent from 2f this is not reflecting detection at all regions.

These data support that H3K4me2 Cut&Tag, shows strong signal at a significant subset of the highly accessible chromatin regions detected by ATAC-seq.

The authors write that H3K4me2 Cut&Tag 'captures accessible chromatin sites in the genome with greater sensitivity' which seems too strong a statement, given that it does not capture all ATAC-seq sites and that the specificity of H3K4me2 Cut&Tag for ATAC-seq sites was not evaluated in this manuscript.

2. Related to Figure 4. In their rebuttal the authors state: "We used MACS2 with a stringent threshold to detect peaks, and signal intensity at these peaks to distinguish NPAT-bound sites from

accessible sites. Of 8689 peaks called on NPAT data that fall outside of the histone gene clusters, 5056 are ATAC sites that are strong enough to exceed the threshold, or ~10% of the ~54,000 previously called ATAC sites."

What are the other 3633 sites that are outside of the histone genes but not in ATAC sites? I don't doubt that most of the peaks outside of the primary NPAT sites fall into annotated ATAC-seq peaks (and similar for CTCF sites). But that's not the same as being able to call open chromatin regions based on the Cut&Tag data alone. The authors need to weaken their statement or present analyses that establish the sensitivity AND specificity for their method to detect open chromatin in addition to the binding profile of the specific factor.

Minor comments:

Figure 2:a The K27me3 ChIP-seq track looks weird – maybe accidentally shifted?

Reviewer #2 (Remarks to the Author):

The authors responded to my questions appropriately, provided additional data and the single cell data and analysis is more convincing now. I have no more comments.

Dr. Gabriella Ficz

Reviewer #1 (Remarks to the Author):

The authors addressed most of my concerns, however there are some points left that should be clarified before acceptance.

1. The analysis shown in figures 2f and 2h is still not clear to me. ATAC-seq identifies open chromatin regions and the authors use ATAC-seq peak calls to compare K4me2 Cut&Tag signal. They state that H3K4me2 detects open chromatin more 'effective' (rebuttal letter) or with 'greater sensitivity' (revised manuscript) than ATAC-seq. This does not seem to be supported by the data presented: The site-by-site comparisons of the heatmaps for ATAC-seq and K4me2 (Fig. 2f) Cut&Tag shows an ATAC-seq signal throughout. This is of course by design, since ATAC-seq peaks were used to select these regions. The K4me2 signal over the same regions on the right side shows that K4me2 shows higher signal at or around ~ 50% of these sites. Figure 2h shows aggregate plots of the data all regions used in 2f and demonstrates that the signal is much higher for K4me2.

Neither of these plots would lead to the conclusion that K4me2 is more effective than ATAC-seq at detecting open chromatin.

1. K4me2 does not seem to detect all open chromatin regions detected by ATAC-seq (2f).

2. The aggregate plot doesn't capture the distribution of the data across all regions. The signal is much higher than for ATAC-seq, but as apparent from 2f this is not reflecting detection at all regions.

These data support that H3K4me2 Cut&Tag, shows strong signal at a significant subset of the highly accessible chromatin regions detected by ATAC-seq.

The aggregate plot in Figure 2h shows that H3K4me2 CUT&Tag has a much higher signal than ATAC-seq. Figure 2f is normalized within each dataset to display the range of signals, and the larger dynamic range of H3K4me2 CUT&Tag means that the display makes it hard to see the lower range of sites without oversaturating the high signals. This is why the K4me2 heat map appears to miss the lower range of accessible sites. We had added the aggregate plot to respond to the concern of Reviewer 1 that the heat map of normalized counts was not quantitative, and this panel shows that without normalization, CUT&Tag has a much higher signal. We have clarified this point in the manuscript (p. 6, first paragraph).

The authors write that H3K4me2 Cut&Tag 'captures accessible chromatin sites in the genome with greater sensitivity' which seems too strong a statement, given that it does not capture all ATAC-seq sites and that the specificity of H3K4me2 Cut&Tag for ATAC-seq sites was not evaluated in this manuscript.

We have toned down the statement to read: 'captures the most prominent accessible chromatin sites in the genome with greater sensitivity.' (p. 6, first paragraph).

2. Related to Figure 4. In their rebuttal the authors state: "We used MACS2 with a stringent threshold to detect peaks, and signal intensity at these peaks to distinguish

NPAT-bound sites from accessible sites. Of 8689 peaks called on NPAT data that fall outside of the histone gene clusters, 5056 are ATAC sites that are strong enough to exceed the threshold, or ~10% of the ~54,000 previously called ATAC sites."

What are the other 3633 sites that are outside of the histone genes but not in ATAC sites? I don't doubt that most of the peaks outside of the primary NPAT sites fall into annotated ATAC-seq peaks (and similar for CTCF sites). But that's not the same as being able to call open chromatin regions based on the Cut&Tag data alone. The authors need to weaken their statement or present analyses that establish the sensitivity AND specificity for their method to detect open chromatin in addition to the binding profile of the specific factor.

The data in Figure 4b-c show NPAT CUT&Tag read counts at annotated histone locus sites (blue) and peaks called from ATAC-seq data (red). Calling peaks using MACS2 on NPAT data suffers from many low count background sites being called as peaks, thus many of the 3633 sites that are outside of the histone genes and are not ATAC-seq-called sites are likely to be background. We now clarify this problem with peak-calling in the main text, and comment that the difference in read counts in Figure 4c suggest that it may be addressed by thresholding signals (p6, last paragraph).

Minor comments:

Figure 2:a The K27me3 ChIP-seq track looks weird – maybe accidentally shifted?
The figure is accurate. The downsampled ChIP-seq track has a background of singlet reads, and on the scale that captures the dynamic range of this track (0-1.8 for 8 million reads) there is a "lawn" of single background fragments. The ENCODE project typically sequences to a depth of 30-50 million reads so that the peaks build up over background (on the 0-8.3 scale of the 50 million read ChIP-seq track the appearance of singlet reads is diminished). To avoid this confusion we now make this point in the legend to Figure 2a.

Reviewer #2 (Remarks to the Author):

The authors responded to my questions appropriately, provided additional data and the single cell data and analysis is more convincing now. I have no more comments.

Dr. Gabriella Ficz